

# Bulk entanglement entropy for photons and gravitons in AdS₃

**Alexandre Belin**[1*]**, Nabil Iqbal**[2†] **and Jorrit Kruthoff** [3§]

**1** CERN, Theory Division, 1 Esplanade des Particules, Geneva 23, CH-1211, Switzerland
**2** Centre for Particle Theory, Department of Mathematical Sciences,
Durham University, South Road, Durham DH1 3LE, UK
**3** Stanford Institute for Theoretical Physics, Stanford University, Stanford, CA 94305, USA

★ a.belin@cern.ch
† nabil.iqbal@durham.ac.uk
§ kruthoff@stanford.edu

## Abstract

We study quantum corrections to holographic entanglement entropy in AdS₃/CFT₂; these are given by the bulk entanglement entropy across the Ryu-Takayanagi surface for all fields in the effective gravitational theory. We consider bulk $U(1)$ gauge fields and gravitons, whose dynamics in AdS₃ are governed by Chern-Simons terms and are therefore topological. In this case the relevant Hilbert space is that of the edge excitations. A novelty of the holographic construction is that such modes live not only on the bulk entanglement cut but also on the AdS boundary. We describe the interplay of these excitations and provide an explicit map to the appropriate extended Hilbert space. We compute the bulk entanglement entropy for the CFT vacuum state and find that the effect of the bulk entanglement entropy is to renormalize the relation between the effective holographic central charge and Newton's constant. We also consider excited states obtained by acting with the $U(1)$ current on the vacuum, and compute the difference in bulk entanglement entropy between these states and the vacuum. We compute this UV-finite difference both in the bulk and in the CFT finding a perfect agreement.

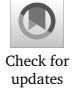
# 1   Introduction

Understanding the entanglement structure of quantum systems has led to profound results in many areas of physics, going from the characterization of topological phases of matter [1,2], to monoticity theorems for the central charges [3–5] and proofs of energy conditions [6,7] in quantum field theory. Perhaps more surprisingly, entanglement has also played a prominent role in elucidating the emergence of spacetime in holography and quantum gravity. This was pioneered by the discovery of the Ryu-Takayanagi (RT) formula for the CFT entanglement entropy [8] in terms of the area of a minimal surface extending into the bulk.

The RT prescription holds at the classical level in the bulk, i.e. to leading order in the large $N$ expansion in the boundary. The quantum corrections were worked out by Faulkner, Lewkowycz and Maldacena (FLM) and read [9]

$$S_{\text{EE}}^{\text{CFT}}(A) = \frac{\text{Area}(\gamma_A)}{4G_N} + S_{\text{EE}}^{\text{bulk}}(\Sigma_A),\qquad(1.1)$$

where $A$ is a boundary subregion, $\gamma_A$ the RT surface and $\Sigma_A$ is the region extending between $\gamma_A$ and $A$. $S_{EE}^{\text{bulk}}$ is the entanglement entropy of all fields present in the bulk effective field theory. Note that the bulk entanglement entropy is UV-divergent; the physics behind this UV divergence is essentially the same as the running of $G_N$, and the $G_N$ appearing in the formula above is the running gravitational constant at the scale of interest. Contributions from one term can shift to the other under the RG flow, and the only unambiguous and UV-finite object is the sum of these two terms.

We are thus led to study the bulk entanglement entropy of the fields that make up the effective theory in the gravitational bulk. Apart from the UV issues that are present for any type of bulk field, there are additional subtleties which will be the object of this work: entanglement of the gauge fields. The bulk effective field theory always contains the graviton, and every continuous global symmetry of the boundary field theory (e.g. the CFT $R$-symmetry) results in a gauge field in the bulk. It is therefore important to understand how to compute the entanglement entropy of such fields. For ordinary gauge fields, there is by now a rich literature on the subject, see e.g. [10–16]. For gravitons much less is known, although there have been discussions about factorizability at the level of the classical phase space [17–20]. A computation of entanglement entropy for massless spin two fields across a sphere was also performed in [21].

Subtleties arise in this context because for gauge fields, the Hilbert space does not factorize between two subregions, even on the lattice. One must therefore be extra careful when cutting open spatial regions. To deal with this issue, it is common to introduce an *extended* Hilbert space [22], such that the Hilbert space of the total system can be embedded into a factorized product

$$\mathcal{H} \subset \mathcal{H}_A \otimes \mathcal{H}_B. \tag{1.2}$$

This procedure must of course be done in a gauge-invariant way, which typically introduces new degrees of freedom at the cut, known as edge modes [10, 13, 23]. The issue is somewhat more severe for certain "ungappable" gauge theories, e.g. Abelian Chern-Simons theory of a single gauge field in three dimensions. Such ungappable gauge theories are often chiral, though this is not a necessary condition [24, 25]. When such gauge theories are placed on a manifold with a physical boundary, the boundary supports gapless edge modes. Relatedly, if we make an *entanglement* cut in order to compute an entanglement entropy, the "same" gapless modes make an appearance at the non-physical entangling surface, as a particular realization of the edge degrees of freedom required to restore gauge invariance. In this case one can imagine that the entangling edge degrees of freedom are gapless, and the powerful techniques of conformal field theory can be used to understand their contribution to the entanglement entropy [26].

In this work, we will study these issues in the context of holography, i.e. we will discuss the bulk gauge theories that arise in examples of $\mathrm{AdS}_3/\mathrm{CFT}_2$. In the simplest case of a boundary $U(1)$ symmetry, the dominant term in the bulk low energy effective action is generally a single Chern-Simons term, resulting in a bulk topological theory. Our task is therefore to compute entanglement entropy in Chern-Simons theory on a manifold with a boundary, when the entanglement cut intersects the boundary; this follows from the FLM prescription. To the best of our knowledge, such a geometry has not been considered before in the rich literature on entanglement entropy in Chern-Simons theory (see for example [27–31] and references therein). We will study this issue carefully and describe the interaction of the modes living on the fictitious entanglement cut with the modes living on the actual physical boundary of the system.

A further application of these issues is to the metric itself, i.e. to quantum gravity. The issue of the factorizability of the quantum gravity Hilbert space is an important open problem. In this work we make some extremely preliminary steps in this direction. In particular, in three bulk dimensions the gravitational theory is topological, and is formally similar to the Chern-Simons theories discussed above; in paticular, the Hilbert space of perturbative excitations is formed from "boundary gravitons", i.e. modes living on the physical boundary [32]. A recent clear exposition of this point can be found in [33].

While most of our analysis is motivated by addressing the question of entanglement in holography, the procedure we discuss is somewhat more general and probes the issue of factorizability of the Hilbert space. We believe our results may have applications to topological phases in condensed matter theory. In particular, our work addresses in the context of a gapless edge theory issues similar to those discussed for a *gapped* edge theory in [34, 35].

**Summary of Results**

In this paper, we provide a construction to cut open the bulk spatial slice in order to write a reduced density matrix, see figure 1. We give an explicit map to the extended Hilbert space which is suitable for topological bulk theories. The map is of the form

$$\mathcal{M} : \mathcal{H} \to \mathcal{H} \otimes \mathcal{H}, \quad |\psi\rangle \to |\psi_E\rangle = \sum_{ij} c_{ij} |E_i\rangle \otimes |E_j\rangle, \tag{1.3}$$

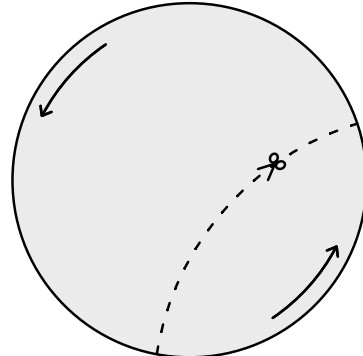
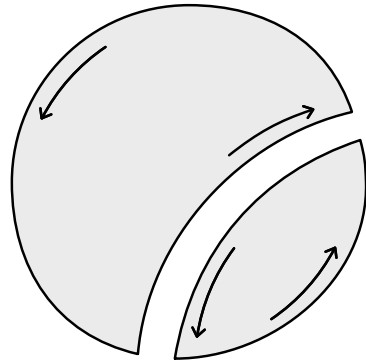

Figure 1: We cut the bulk spatial slice open along the RT surface and factorise the bulk Hilbert space explicitly. The edge modes now run both along the bulk cut and the boundary of $AdS_3$.

for coefficients $c_{ij}$ that we compute and which depend on the original choice of state for the full theory. As we will explain, this computation is simplest when the UV regulator is picked such that entanglement cut degrees of freedom and the physical boundary degrees of freedom are the same, along with a *transparent* boundary condition at the junction; relaxing these assumptions requires a more involved computation (though the methodology we propose still applies). The state (1.3) can be viewed as a generalization of a high temperature thermal state, where the temperature plays the role of the inverse cutoff.

Given this choice of regulator, we can compute the bulk entanglement entropy for a generic topological field theory in the bulk. For a boundary interval of angle $\theta$ we find in the vacuum

$$S_{\text{bulk}} = \frac{c_{\text{top}}}{3} \log\left(\frac{2}{\epsilon_{\text{CFT}}} \sin\left(\frac{\theta}{2}\right)\right) + \frac{c_{\text{top}}}{3} \frac{L}{2\epsilon_{\text{bulk}}} + S_{\text{top}}, \qquad (1.4)$$

where $L$ is the length of the *bulk* entanglement cut, and where $\epsilon_{\text{bulk}}, \epsilon_{\text{CFT}}$ are the bulk and boundary UV-cutoffs, and $c_{\text{top}}$ is a number parametrizing the number of edge degrees of freedom. The result presented above is valid for any topological theory placed on a spatial disk, when the entanglement cut intersects the boundary. In holography, the FLM relation (1.1) implies that $L$ is fixed to be the length of the bulk geodesic, and combining this with the classical RT result we find the following dual CFT entropy:[1]

$$S_{\text{CFT}} = \frac{A}{4G_N} + S_{\text{bulk}} = \left(\frac{\ell_{\text{AdS}}}{2G_N} + \frac{c_{\text{top}}}{3} \frac{\ell_{\text{AdS}}}{\epsilon_{\text{bulk}}} + \frac{c_{\text{top}}}{3}\right) \log\left(\frac{2}{\epsilon_{\text{CFT}}} \sin\left(\frac{\theta}{2}\right)\right). \qquad (1.5)$$

We find that the bulk entanglement entropy is a sum of two terms, both of which effectively renormalize the relation between the central charge of the holographic CFT and $G_N$. Symmetry arguments imply that it was the only possible consistent outcome, since the entanglement entropy of an interval in the vacuum is fixed by symmetry. Our result thus illustrates this phenomenon.

The above framework is completely general for any topological field theory in the bulk, and so applies straightforwardly to the case of a $U(1)$ Chern-Simons theory, which is our main application. We also boldly apply it to the case of the 3d graviton, where (given certain assumptions) we also find reasonable results. (Here $c_{\text{top}} = \frac{1}{2}$ for a single chiral $U(1)$ and $c_{\text{top}} = 1$ for left- and right-moving boundary gravitons).

In the case of a $U(1)$ gauge theory we also go further, computing the change in the entanglement entropy for excited states. The bulk theory is topological so the area does not change

---

[1] We will drop the constant piece $S_{\text{top}}$ since it is not universal and can be changed by rescaling the CFT cutoff.

but we compute the difference in bulk entanglement entropy. We find complete agreement with the CFT answer, therefore providing a check of the FLM formula. To the best of our knowledge, this is the only check of the FLM formula (1.1) which is not fixed by conformal symmetry and does not require an expansion, holding for arbitrary interval size.

The paper is organised as follows. We start in section 2 by a brief discussion of the CFT computation of entanglement entropy in 2$d$ CFTs. We discuss excited states, in particular the state where we act with a $U(1)$ conserved current on the vacuum. In section 3 we move to the main part of the paper and discuss the computation of the bulk entanglement entropy. This requires splitting the bulk Hilbert space which we discuss in detail. We then apply the resulting procedure to $U(1)$ Chern-Simons theory. In section 4, we discuss the bulk entanglement entropy for the boundary gravitons. We conclude in section 5 with various extensions of our computations and an interpretation for the renormalization of $G_N$. In appendix A we collect some known results about Chern-Simons theory and discuss the case where the theory has both left and right-moving sectors leading to a non-chiral boson along its boundary. We compute the vacuum entanglement from the $U(1)$ Chern-Simons wave functional on the torus in appendix B.

In the final stages of preparation of this paper, [36] appeared, which numerically computes the entanglement entropy in integer quantum hall states with an entanglement cut intersecting a physical boundary; their results agree with our EFT approach where a comparison is possible. [37] also appeared, where the entanglement entropy in Jackiw-Teitelboim gravity is computed; this is a detailed two-dimensional counterpart of the three-dimensional calculation outlined in section 4.

## 2 CFT calculation

In this section, we give a short review of the method to compute entanglement entropy in 2d CFTs, with a focus on excited states of large $c$ CFTs. For a more in depth review of the subject, we refer the reader to [38–41].

### 2.1 Entanglement in CFT$_2$

Consider a 2d CFT in a state $|\psi\rangle$. We will divide the Hilbert space into two subsystems, $A$ and its complement $\bar{A}$. The reduced density matrix of the subsystem $A$ is given by

$$\rho_A \equiv \text{Tr}_{\bar{A}} |\psi\rangle \langle \psi|, \tag{2.1}$$

from which we can compute the entanglement entropy, which is the von Neumann entropy of the reduced density matrix

$$S_{\text{EE}} = -\text{Tr}\rho_A \log \rho_A. \tag{2.2}$$

In quantum field theory, it is often difficult to directly compute the entanglement entropy so it is common to resort to the replica trick [38,42]. We first compute the Rényi entropies

$$S_n \equiv \frac{1}{1-n} \log \text{Tr}\rho_A^n, \tag{2.3}$$

and then analytically continue the Rényi entropies in $n$ to obtain the entanglement entropy:[2]

$$S_{\text{EE}} = \lim_{n \to 1} S_n. \tag{2.4}$$

---

[2]At large central charge, subtleties can appear in the analytic continuation [43–47]. To the best of our knowledge, they do not play any role for the type of states discussed here.

In this paper, we will consider a 2d CFT $\mathcal{C}$ on a circle of length $2\pi$ parametrized by a coordinate $\varphi$ and we define the subsytem $A$ to be the spatial interval with size $\theta$.

The states we will be interested in are those obtained by acting with a primary operator on the vacuum, namely

$$|\psi\rangle = O(0)|0\rangle, \tag{2.5}$$

for a Virasoro primary operator $O$ with dimensions $(h, \bar{h})$. The dual state is given by

$$\langle\psi| = \lim_{z \to \infty} \langle 0| O(z) z^{2h} \bar{z}^{2\bar{h}}. \tag{2.6}$$

When the operator is the identity, namely the state is the CFT vacuum, the entanglement entropy is fixed by symmetry and reads

$$S_{\text{EE}} = \frac{c}{3} \log\left( \frac{2}{\epsilon_{\text{CFT}}} \sin\left(\frac{\theta}{2}\right) \right), \tag{2.7}$$

which is UV-divergent and we have introduced a CFT cutoff $\epsilon_{\text{CFT}}$.

We will be interested in computing a UV-finite quantity which is the difference in entanglement entropies between an excited state and the vacuum. We start by computing the difference in Rényi entropies

$$\Delta S_n \equiv S_n^{\text{ex}} - S_n^{\text{vac}} = \frac{1}{1-n} \log \frac{\text{Tr}\rho_A^n}{\text{Tr}\rho_{A,\text{vac}}^n}. \tag{2.8}$$

After a conformal transformation, one can map this quantity to $2n$-point correlation function on the plane [39].

$$\frac{\text{Tr}\rho_A^n}{\text{Tr}\rho_{A,\text{vac}}^n} = e^{-i\theta(h-\bar{h})} \left( \frac{2}{n} \sin\left[\frac{\theta}{2}\right] \right)^{2n(h+\bar{h})} \left\langle \prod_{k=0}^{n-1} O(\tilde{z}_k) O(z_k) \right\rangle, \tag{2.9}$$

with

$$z_k = e^{-i(\theta - 2\pi k)/n}, \qquad \tilde{z}_k = e^{2\pi i k/n}, \qquad k = 0, ..., n-1. \tag{2.10}$$

Once the conformal transformation has been performed, the $2n$ operators lie on the unit circle, which we will call the *clock* geometry, see figure 2. Equation (2.9) is the key formula to compute the entanglement entropy for the excited state.

So far, we have reviewed the general procedure that works for all primary states of 2d CFTs. We will now focus on large $c$ CFTs and in particular on the excited state obtained by acting with a global $U(1)$ current on the vacuum.

## 2.2 Current states

In the first part of this work, we will assume that the CFT possesses a global $U(1)$ symmetry such that the chiral algebra is given by a $U(1)$ Kac-Moody algebra at level $k$. As we have seen, the Rényi entropies of certain excited states are given by $2n$-point correlation functions which in general are hard to compute. Fortunately, we are interested in states created by the insertion of a $U(1)$ current and the arbitrary point correlation function of currents is completely fixed by symmetry! While large $c$ (in particular large $c$ factorization) was a crucial ingredient to have control over the Rényi entropies for scalar excitations [41], it is somewhat less crucial here since the $2n$-point function is fixed by a symmetry independently of the value of $k$. Our calculations will not depend on the value of $k$, but in applying our results to AdS/CFT following the RT and FLM prescriptions, we should of course take the level $k$ to be large.[3]

---

[3] In the microscopic examples of $AdS_3/CFT_2$ such as the D1D5 CFT, the level is related to the central charge by supersymmetry and we have $k = c/6$. In that case, the global symmetry is actually bigger, namely $SU(2)$, and we would take the $U(1)$ to be in the Cartan subalgebra.



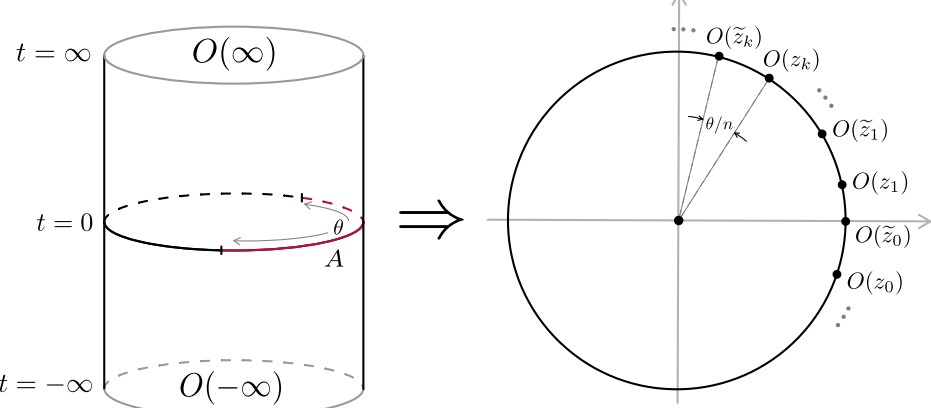

Figure 2: Geometry of our set-up. *Left*: The geometry before the uniformization map. The operators $O$ are inserted at $\pm\infty$ to create $|\psi\rangle$ and $\langle\psi|$. The region $A$ of length $\theta$ is displayed in red. *Right*: Clockwise arrangement of operators along the unit circle in the replicated theory after unformization and a conformal map to the plane. The distance between the operators at $z_k$ and $\widetilde{z}_k$ along the unit circle is $\theta/n$, the interval length divided by $n$.

It is straightforward to compute the correlation function (2.9). One essentially computes all possible Wick contractions of the current. At that level, the calculation resembles that of the scalar excitation, although in this case it is exact. It is important to note that we have normalised the $U(1)$ current $J$ so that $J(z)J(0) \sim k/z^2$. This matches the canonical normalisation of the Abelian Chern-Simons action we discuss below. This normalisation will not enter in the difference in Rényi entropies. They read,

$$\Delta S_n = \frac{1}{1-n} \log\left(\left(\frac{1}{n}\sin\left[\frac{\theta}{2}\right]\right)^{2n} \mathrm{Hf}(M_{ij})\right), \tag{2.11}$$

where $\mathrm{Hf}(M)$ is the Haffnian of a matrix $M$ defined by

$$\mathrm{Hf}(M) = \frac{1}{2^n n!} \sum_{g \in S_{2n}} \prod_{j=1}^{n} M_{g(2j-1),g(2j)}, \tag{2.12}$$

and

$$M_{ij} = \begin{cases} \frac{1}{(\sin\frac{\pi(i-j)}{n})^2}, & i,j \leq n \\ \frac{1}{\left(\sin\left(\frac{\pi(i-j)}{n} - \frac{\theta}{2n}\right)\right)^2}, & i \leq n, \quad j > n \\ \frac{1}{\left(\sin\left(\frac{\pi(i-j)}{n} + \frac{\theta}{2n}\right)\right)^2}, & j \leq n, \quad i > n \\ \frac{1}{(\sin\frac{\pi(i-j)}{n})^2}, & i,j > n. \end{cases} \tag{2.13}$$

This is the exact expression for the difference in Rényi entropy of a current state. From this, one can actually perform the analytic continuation and obtain the final answer for the change in the entanglement entropy [48–51]

$$\Delta S_{\mathrm{EE}} = S_{\mathrm{EE}}^{\mathrm{current}} - S_{\mathrm{EE}}^{\mathrm{vac}} = -2\left(\log\left(2\sin\frac{\theta}{2}\right) + \Psi\left(\frac{1}{2\sin\frac{\theta}{2}}\right) + \sin\frac{\theta}{2}\right). \tag{2.14}$$

One of the goals of this work is to reproduce this answer from the FLM formula in the dual bulk theory, to which we now turn.

# 3  Bulk entanglement entropy for photons

In this section we discuss the bulk computation. Including quantum corrections, the entanglement entropy of the boundary is given in the bulk by the FLM formula [9]

$$S_{\text{EE}}^{\text{CFT}}(A) = \frac{\text{Area}(\gamma_A)}{4G_N} + S_{\text{EE}}^{\text{bulk}}(\Sigma_A), \tag{3.1}$$

where the bulk region $\Sigma_A$ is the region between the boundary region $A$ and the bulk minimal surface $\gamma_A$. The bulk entanglement entropy piece is the Von Neumann entropy of the bulk matter fields in the bulk quantum state. In this section, we study the bulk dynamics dual to a global $U(1)$ current on the boundary. Unlike in higher dimensional AdS/CFT, the low energy effective theory of the bulk $U(1)$ gauge field is not given by a Maxwell term, but rather by a Chern-Simons term [52]. The low-energy effective theory for the photons is therefore a Chern-Simons theory, which is topological.[4] It turns out that specifying *exactly* which low-energy theory governs the bulk dynamics is a slightly subtle question that we will address below. For this reason, we start by reviewing the salient features of $U(1)$ Chern-Simons theory that are relevant for this work. We will be rather brief here, but see Appendix A for more details and references.

## 3.1  Chern-Simons theory and the chiral boson

To begin, it is well-understood that the holomorphic sector of a $U(1)$ Kac-Moody algebra is represented holographically by $U(1)$ Chern-Simons theory on $\text{AdS}_3$:

$$S_{CS}[A] = \frac{k}{4\pi} \int A \wedge dA = \frac{k}{4\pi} \int d^3x \, \epsilon^{\mu\nu\rho} A_\mu \partial_\nu A_\rho, \tag{3.2}$$

$$A \to A + d\Lambda, \qquad \Lambda \sim \Lambda + 2\pi, \tag{3.3}$$

with unit electric charges that couple to the gauge field as $\exp(i \int A)$. The Dirac condition implies that over all closed 2-manifolds, $\int_{\mathcal{M}^2} dA = 2\pi\mathbb{Z}$. $k$ is an integer that maps to the level of the dual Kac-Moody algebra. There is a subtle difference between even and odd $k$, depending on the three-manifold one considers. If the three-manifold is a non-spin manifold, then $k$ has to be even, whereas for spin manifolds, $k$ can be odd, provided one also chooses a particular spin-structure. See [55] for a beautiful explanation of this subtle difference between even and odd $k$.

   We will study this theory on a manifold with boundary. As is well-known, the CS theory itself has no local dynamics but acquires a propagating chiral boson edge mode $\phi(z)$ at level $k$ in the presence of a boundary. This edge theory is purely holomorphic. In what follows it will be important to understand the operator content of the edge theory exactly. There are two main players:

i) The theory contains a single holomorphic current $j(z) = k\partial\phi$ whose modes form a $U(1)$ Kac-Moody algebra. This algebra has level $k$. In the Abelian case this statement only has meaning once we define the charge quantization conditions. These conditions are inherited from the charge quantization condition of the compactness of the gauge

---

[4]There will of course also generically be non-topological higher derivative corrections present in the bulk, e.g. the quadratic Maxwell itself. As the bulk entangling region reaches the AdS boundary, we are discussing an extremely infrared observable from the bulk point of view, and need not consider the effects of such terms. We note that in the context of entanglement entropy in flat space the interplay between Maxwell and Chern-Simons terms has been studied in [53, 54], resulting in a crossover at intermediate scales; though not relevant for the specific observable we study, it would be interesting to understand similar issues in the AdS context.

group. From the bulk point of view, the modes of this Kac-Moody algebra are "boundary photons".

ii) The theory also contains a *chiral vertex operator*, i.e. a purely holomorphic operator with $U(1)$ charge $k$. Its holomorphic dimension is $h = \frac{k}{2}$; note that this is allowed to be half-integer, meaning that if $k$ is odd, it is a fermionic operator. In our normalization, these states are given by the operator $e^{ik\phi(z)}$, where the argument of $\phi$ indicates that only the holomorphic part of $\phi(z)$ is taken. In a small vulgarization we can say that from the bulk point of view, this state is a "Dirac string".

We emphasize that this operator algebra consists only of states formed out of the bulk photon itself; as they are all purely holomorphic with (half-)integer dimensions, they should be thought of as constituting an enlarged symmetry algebra. In particular, states corresponding to Wilson lines are *not* operators in this chiral operator algebra.[5]

Consequently, whenever we cut the 3d Chern-Simons theory, the exposed 2d surface acquires a CFT "skin". We will call this $\mathrm{CFT}_B$, and will refer to it as the *boundary* CFT. In the example discussed above, the properties of this CFT are universal.[6] We should stress that this boundary CFT captures part of the physical excitation spectrum of the Chern-Simons theory, and so is not *dual* to it, just as the skin of an orange is not dual to the orange.

Note that the low-lying Hilbert space about the vacuum of $\mathrm{AdS}_3$ is thus made entirely from degrees of freedom in $\mathrm{CFT}_B$, which is defined on the boundary circle. We denote this Hilbert space by $\mathcal{H}^B$.

## 3.2 Factorization of the bulk Hilbert space

We now finally turn to the entanglement entropy of a region $A$ in the dual field theory. In the usual fashion we are instructed to study a geodesic $\gamma_A$ hanging down into the bulk; we denote the region between the geodesic and the boundary by $\Sigma_A$, see figure 3. The FLM prescription (3.1) tells us that the full entanglement entropy in the boundary theory is the area piece plus a bulk piece $S_{\mathrm{bulk}}(\Sigma_A)$.

To compute $S_{\mathrm{bulk}}$ we need to cut the bulk theory along the entangling surface $\gamma_A$, i.e. we need to make an *entanglement* cut in a Chern-Simons theory. We are however unable to factorize the bulk Hilbert space without introducing extra degrees of freedom; in other words, as discussed above, this entanglement cut will also acquire a CFT "skin", which we call $\mathrm{CFT}_E$.

The properties of this entanglement cut are associated with details of the UV completion of the bulk theory. These considerations are distinct from those determining the boundary CFT, and thus in general the entanglement cut $\mathrm{CFT}_E$ is distinct from the boundary mode $\mathrm{CFT}_B$, though some aspects will be universal. The details depend on the precise theory under consideration. One possibility is that they are related by RG flow; for example a relevant deformation could be turned on for $\mathrm{CFT}_E$ and not for $\mathrm{CFT}_B$. It is sometimes (though certainly not always) even possible to gap out $\mathrm{CFT}_E$ entirely. We will discuss some examples of this sort below. In all situations, the combined Hilbert space of the factorized theory is

$$\mathcal{H}_{\mathrm{factorized}} = \mathcal{H}^{\mathrm{tot}}_{\partial\Sigma_A} \otimes \mathcal{H}^{\mathrm{tot}}_{\partial\Sigma_{\bar{A}}}, \tag{3.4}$$

where $\partial\Sigma_A = \gamma_A \cup A$ and $\mathcal{H}^{\mathrm{tot}}_{\partial\Sigma_A}$ is the total Hilbert space of $\mathrm{CFT}_E$ and $\mathrm{CFT}_B$. Note that these two Hilbert spaces generically will not factorize between $\gamma_A$ and $A$. Similar notation is used for the complementary regions.

---

[5]In an application to holography, such Wilson lines will correspond to the massive quanta of bulk charged fields that are dual to other primary operators outside the chiral algebra, and are thus not described by the Chern-Simons theory alone.

[6]This is not necessarily the case; for example, if we broke boundary Lorentz-invariance, the speed of the propagating mode would not be fixed by the bulk theory and would be a non-universal tunable parameter.

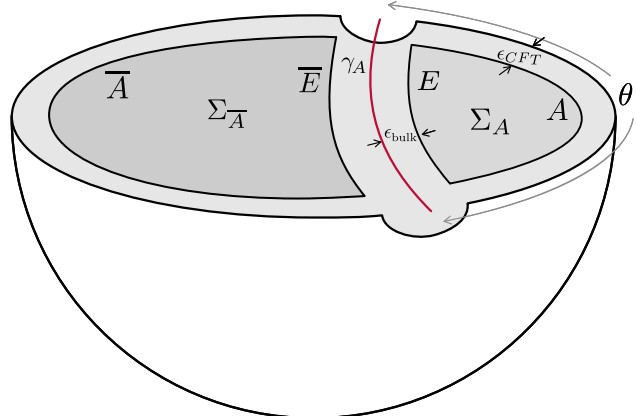

Figure 3: Bulk geometry. The complementary bulk regions are indicated by $\Sigma_A$ and $\Sigma_{\bar{A}}$ and their corresponding boundary regions by $A, \bar{A}$. The angular size of $A$ is $\theta$. The edges on either side of the entanglement cut along the RT surface $\gamma_A$ are indicated by $E$ and $\bar{E}$. The bulk cutoff is $\epsilon_{\text{bulk}}$ and the boundary CFT cutoff $\epsilon_{\text{CFT}}$.

Our task is now to determine how a given initial state in the Hilbert space of $\mathcal{H}^B$ is embedded into this larger Hilbert space. The two regions $\gamma_{\bar{A}} \cup \bar{A}$ and $\gamma_A \cup A$ are both individually topologically $S^1$, and the Hilbert space is a genuine tensor product across these two $S^1$'s. We can then trace out one of these $S^1$'s to obtain a reduced density matrix from which we can compute the entanglement entropy.

Now, as mentioned above the two theories $\text{CFT}_E$ and $\text{CFT}_B$ are generically not the same theory. We thus have to describe the interface at the junction between the entanglement cut and the boundary. The problem of how to glue together two CFTs along a conformal interface is well studied (see e.g. [56, 57]); in the generic case this technology could be used to attack this problem.

For our purposes, a particularly convenient case is when $\text{CFT}_E$ is the *same* as $\text{CFT}_B$, and moreover when the interface between them is perfectly transparent. In this case (3.4) becomes

$$\mathcal{H}_{\text{factorized}} = \mathcal{H}^B_{\gamma_{\bar{A}} \cup \bar{A}} \otimes \mathcal{H}^B_{\gamma_A \cup A}. \tag{3.5}$$

Whether or not this possibility is actually realized will depend on the details of the UV regularization, but we will begin our discussion assuming it to be the case. Given this choice of junction condition, we now discuss how to map arbitrary states in $\mathcal{H}^B$ to $\mathcal{H}_{\text{factorized}}$.

### 3.3 Conformal transformations

We are considering the entanglement entropy of an interval of length $\theta$ on the boundary cylinder. We would like to construct a 2d surface that connects two small discs of radius $\epsilon_{\text{CFT}}$ (each surrounding one of the endpoints of the interval) on the boundary by a long and narrow tube that goes through the interior. This is represented in figure 3. In applications to holography, this tube should follow a bulk geodesic. We take the radius of the interior tube to be $\epsilon_{\text{bulk}}$, and denote its length by $L$. This surface is topologically a torus; as the "skin" theory is conformal, it cares only about the modular parameter of this torus, which must be some function of the data $\theta, \epsilon_{\text{CFT}}, \epsilon_{\text{bulk}}$, and $L$. We now calculate this modular parameter by constructing the conformal transformation that maps this complicated shape to a canonical torus. We first cut out a circle of radius $\epsilon_F$ around $z = 0$ and $z = \theta$, where $\epsilon_F < \epsilon_{\text{CFT}}$, and then identify these two circles. The region where $\epsilon_F < |z| < \epsilon_{\text{CFT}}$ together with its counterpart $\epsilon_F < |z - \theta| < \epsilon_{\text{CFT}}$ are glued together along $\epsilon_F$ and form the bulk tube, see figure 4. We now

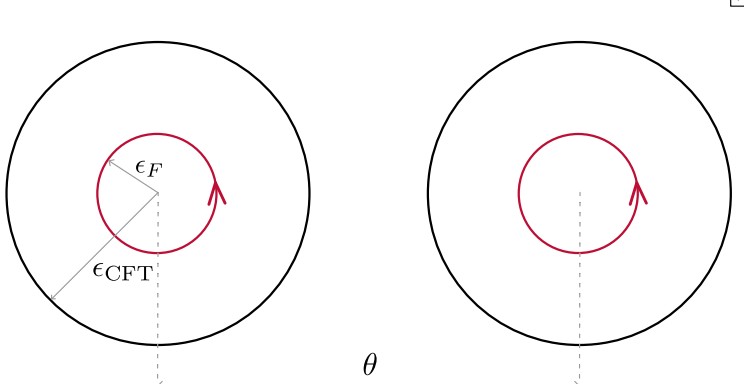

Figure 4: The bulk geometry in the $z$-plane. The two red circles with radius $\epsilon_F$ are identified, whereas the black circles of radius $\epsilon_{\mathrm{CFT}}$ represent the CFT cutoff. The distance between the two circles is $\theta$, the size of the interval on the boundary. Once glued together the region in between the two black circles represents the bulk tube around the entanglement cut.

map to a coordinate $w$ that is naturally aligned with this tube; in particular, the mapping that we use is

$$\frac{\sin\left(\frac{z}{2}\right)}{\sin\left(\frac{z-\theta}{2}\right)} = \exp\left(\frac{w}{\epsilon_{\mathrm{bulk}}}\right), \tag{3.6}$$

where $z$ is the coordinate on the cylinder which is $z = \varphi + i\tau$, with $\varphi \sim \varphi + 2\pi$, and where $\theta$ is the size of the interval.

We see that $w$ has imaginary periodicity $2\pi\epsilon_{\mathrm{bulk}}$ and the real range of $w$ is found by moving from the circle at $z = \epsilon_F$ to the circle at $z - \theta = \epsilon_F$.[7] We can solve this to find the range of $w$

$$\frac{1}{2}\frac{\epsilon_F}{\sin\left(\frac{\theta}{2}\right)} = \exp\left(\frac{w_1}{\epsilon_{\mathrm{bulk}}}\right), \qquad \frac{1}{2}\frac{\epsilon_F}{\sin\left(\frac{\theta}{2}\right)} = \exp\left(-\frac{w_2}{\epsilon_{\mathrm{bulk}}}\right). \tag{3.7}$$

The difference between $w_1$ and $w_2$ is then

$$w_2 - w_1 = 2\epsilon_{\mathrm{bulk}}\log\left(\frac{2}{\epsilon_F}\sin\left(\frac{\theta}{2}\right)\right). \tag{3.8}$$

Thus the $w$ coordinate is a torus with the two cycles having length $2\epsilon_{\mathrm{bulk}}\log\left(\frac{2}{\epsilon_F}\sin\left(\frac{\theta}{2}\right)\right)$ and $2\pi\epsilon_{\mathrm{bulk}}$.

Now, part of the $w$ torus consists of the boundary cylinder and part of it is the bulk tube, where the dividing line between them is the circle at radius $\epsilon_{\mathrm{CFT}}$. Thus the separation in $w$ which measures the length of the bulk tube is

$$L = \Delta w = 2\epsilon_{\mathrm{bulk}}\log\left(\frac{\epsilon_{\mathrm{CFT}}}{\epsilon_F}\right). \tag{3.9}$$

This expression should be viewed as a way to find the fictitious parameter $\epsilon_F$ as a function of $\epsilon_{\mathrm{CFT}}, L$, and $\epsilon_{\mathrm{bulk}}$. We note that $L$ and $\epsilon_{\mathrm{bulk}}$ individually have no meaning; however their conformally invariant ratio does.

---

[7]The torus we obtain from this identification is in general not flat, but it becomes flat to leading order in the small cutoff expansion, and we will work to this order. The higher order corrections can be tracked and only change our results up to terms that vanish as the cutoff is taken to zero, so we will neglect them.

It will be convenient in what follows to perform a rescaling and define a new coordinate as

$$u = \frac{\pi}{\epsilon_{\text{bulk}} \log \frac{2}{\epsilon_F} \sin\left(\frac{\theta}{2}\right)} w, \tag{3.10}$$

with the following identifications

$$u \sim u + 2\pi, \quad u \sim u + i\beta, \qquad \beta = \frac{2\pi^2}{\log\left(\frac{2}{\epsilon_F} \sin\left(\frac{\theta}{2}\right)\right)}, \tag{3.11}$$

and we will take the range of the imaginary part of $u$ to be $[-i\frac{\beta}{2}, i\frac{\beta}{2})$.

Having discussed the bulk geometry that we want to consider, we are now ready to move on to the computation of the bulk entanglement entropy in both the vacuum and an excited state.

## 3.4 Torus partition function

We begin by considering the case of the vacuum entanglement entropy, i.e. we study the partition function $Z(\beta)$ on the torus given by the identification pattern (3.11). We first turn to a precise specification of what we mean by $Z(\beta)$. In fact the theories of interest do not have a partition function but rather a *vector* of partition functions [58]. We must specify which component of this vector we are interested in. A basis for this vector space is provided by operators which are primaries under the extended chiral algebra $\mathcal{A}$ of interest.

For example, consider the basic $U(1)$ Chern-Simons theory as described in Section 3.1. In this case $\mathcal{A}$ generated by the modes of $\partial\phi$ and $e^{ik\phi}$. There are however in principle different choices of *vacuum* that this operator algebra can act on: in particular, we may consider the different vacuua formed by

$$|m\rangle = e^{im\phi}|0\rangle, \tag{3.12}$$

where $|0\rangle$ is the state with zero $U(1)$ charge and with $m = 0, \ldots, k-1$. Denoting the space of states formed by acting with the chiral algebra $\mathcal{A}$ on the vacuum $|m\rangle$ by $\mathcal{H}_m$, the most general partition sum that we can compute is

$$\chi_m(\beta) = \sum_{n \in \mathcal{H}_m} \exp(-\beta E_n). \tag{3.13}$$

This is a character of the extended chiral algebra, labeled by $m$. As we are computing the partition function of the torus with no Wilson lines inserted in the interior [59], we are then interested in the case with zero charge, i.e. $m = 0$.

From (3.11), we see that at small cutoffs, we are interested in the limit $\beta \to 0$, i.e. in the high temperature limit. The temperature thus serves as a UV-cutoff and diverges in the limit where the cutoff vanishes. Within this type of regulator, it is then standard to obtain the entanglement entropy using the Cardy formula [60]. However as we are now dealing with a character and not a modular-invariant partition function, we must take some care in performing the $S$-transform. We find:

$$\chi_0(\beta) = \sum_m S_{0m} \chi_m\left(\frac{4\pi^2}{\beta}\right), \tag{3.14}$$

where the modular S-matrix [61,62] makes an appearance. Now if we take the limit $\beta \to 0$, only the vacuum contribution $\chi_0$ contributes from the sum over characters. We find:

$$\chi_0(\beta \to 0) = S_{00} \chi_0\left(\frac{4\pi^2}{\beta}\right) \approx S_{00} \exp\left(\frac{2\pi^2(c_L + c_R)}{12\beta}\right), \tag{3.15}$$

where we have further kept only the first contribution to the character itself (i.e. the vacuum contribution with energy $E_0 = -(c_L + c_R)/24$) in the sum over states. Computing the entropy as

$$S = \left(1 - \beta \partial_\beta\right) \log \chi_0(\beta), \tag{3.16}$$

and using the formula for $\beta$ in (3.11), we find

$$S = \frac{c_L + c_R}{6} \log\left(\frac{2}{\epsilon_F} \sin\left(\frac{\theta}{2}\right)\right) + \log S_{00}. \tag{3.17}$$

We may now finally specialize to the case of interest, where the theory is the chiral boson at level $k$; in that case we have $(c_L, c_R) = (1, 0)$, and the modular S-matrix is [61]

$$S_{mn} = \frac{1}{\sqrt{k}} \exp\left(-\frac{2\pi i}{k} mn\right). \tag{3.18}$$

We thus find for the entropy in this case

$$S = \frac{1}{6} \log\left(\frac{2}{\epsilon_F} \sin\left(\frac{\theta}{2}\right)\right) - \frac{1}{2} \log k. \tag{3.19}$$

For completeness, in Appendix B we also present an explicit derivation of the same result using the known wavefunctions of the bulk Chern-Simons theory.

Even though we used the Abelian Chern-Simons theory as an example, it should be clear from the generality of the discussion that the result (3.17) applies to any theory with an extended chiral algebra; we must simply use the appropriate $S$ matrix. For example, when a Wilson line is inserted in the bulk, $m \neq 0$ and we simply consider the character $\chi_m$. The computation is then analogous to the one presented above, but $\log S_{00}$ is replaced by $\log S_{m0}$. Note that for a free chiral boson theory, there is no distinction between $S_{m0}$ and $S_{00}$, so the answer remains the same.

Let us now discuss the answer. Using (3.9) to express $\epsilon_F$ in terms of quantities with physical significance, we get

$$S = \frac{1}{6} \log\left(\frac{2}{\epsilon_{\text{CFT}}} \sin\left(\frac{\theta}{2}\right)\right) + \frac{1}{6} \frac{L}{2\epsilon_{\text{bulk}}} - \frac{1}{2} \log k. \tag{3.20}$$

Each term in this expression has a distinct interpretation:

i) The first term arises from the modes living on the physical boundary of the space, i.e. the boundary chiral boson modes. We see that this takes the familiar form of a vacuum entanglement entropy in a CFT with $c_L = 1, c_R = 0$, as befits a chiral boson. In a general Chern-Simons theory (i.e. without considering a holographic interpretation), we could imagine picking $L$ independently from $\theta$, and thus the coefficient of the logarithmic term is clearly universal. The term has the usual $\text{CFT}_2$ dependence on the length of the boundary interval $\theta$ measured in units of the boundary cutoff $\epsilon_{\text{CFT}}$.

ii) The second term can be thought of as arising from the modes living on the bulk entanglement cut. It takes precisely the expected form namely a 3d "area term", i.e. it measures the bulk distance along the cut in units of the bulk UV cutoff.

iii) The final term is associated with the fact that the bulk Chern-Simons theory is topologically ordered. In the usual construction of "topological entanglement entropy" [1, 63] one considers combinations of geometries from which this term can be cleanly extracted. In our calculation however there does not appear to be a simple way to disentangle this from the CFT cutoff-dependence appearing in the first term; its universal character is spoiled by the gapless modes living on the physical entanglement cut.

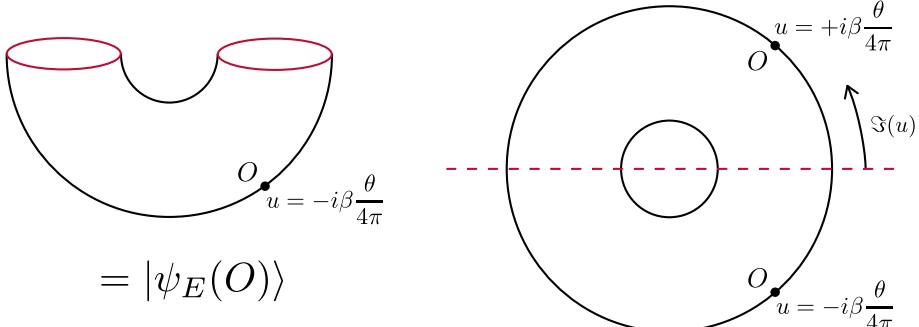

Figure 5: Path integral representation of the state $|\psi_E(O)\rangle$ and its norm in $u$ coordinates. *Left*: Path integral representation of the state $|\psi_E(O)\rangle$ in the extended Hilbert space. *Right*: The norm of $|\psi_E(O)\rangle$. The red dashed line indicates where we cut the $u$-torus open.

The result (3.19) is valid in the general context of $U(1)$ Chern-Simons theory. However when we apply the result to AdS/CFT, we are instructed to take $L$ to be the length of the Ryu-Takayanagi surface, yielding

$$S = \left(\frac{\ell_{\mathrm{AdS}}}{6\epsilon_{\mathrm{bulk}}} + \frac{1}{6}\right) \log\left(\frac{2}{\epsilon_{\mathrm{CFT}}} \sin\left(\frac{\theta}{2}\right)\right), \tag{3.21}$$

where we have dropped the constant term since it is not universal and can be changed by tuning the CFT cutoff. The answer is proportional to the CFT entanglement entropy, as must be the case since the CFT answer is fixed by symmetry. The bulk entanglement entropy therefore only renormalizes Newton's constant. Note that there are two terms, a bulk UV-finite shift by $\frac{1}{6}$ coming from the boundary photons, and a divergent piece coming from the entanglement cut degrees of freedom: however in the holographic context it is not clear whether we can disentangle them. We will return to this in the discussion section.

### 3.5 Excited state and OPE coefficients

In this section, we will provide the details of the map between the original Hilbert space and the extended one. As we have seen, the transparent boundary conditions imply that the map is of the form

$$\mathcal{M} : \mathcal{H} \to \mathcal{H} \otimes \mathcal{H}, \quad |\psi\rangle \to |\psi_E\rangle = \sum_{ij} c_{ij} |E_i\rangle \otimes |E_j\rangle. \tag{3.22}$$

We will now derive the value of the coefficients $c_{ij}$ for an arbitrary state of the boundary CFT Hilbert space. Through the state-operator correspondence, the CFT Hilbert space is given by the set of local operators inserted at the origin of the complex plane, or at $\tau = -i\infty$ in the cylinder coordinates. The full cylinder represents an overlap between the bra and the ket states, which means there is also another operator inserted at $\tau = i\infty$.

After performing the conformal transformation (3.6) and the rescaling, the operators are mapped to $u = \pm i\beta\theta/4\pi$ on the $u$-torus. To obtain a state, one must slice the euclidean path integral open which we will do along the circle $\mathrm{Im}(u) = 0$. The state is now prepared by a euclidean path integral on a cylinder of length $\beta/2$ with the primary operator inserted, as we show in figure 5. One can think of this state as a generalization of the thermofield double state, with an additional operator inserted. The temperature of the TFD-like state is very high and diverges as the cutoff is taken to zero.

It is now quite simple to write down the state in the extended Hilbert space obtained from the original state $O(0)|0\rangle$. To understand the precise nature of the state, we can glue energy

eigenstates on the two open circles of the state $|\psi_E\rangle$. We are therefore computing

$$c_{ij} = \langle E_i| \otimes \langle E_j| \, |\psi_E\rangle. \tag{3.23}$$

To compute this overlap, it is convenient to proceed in the following steps:

1. Start from the state defined on the cylinder, and map it to the plane through the exponential map. It is now a piece of the complex plane, extending from $|z| = 1$ to $|z| = e^{\beta/2}$, where $z$ is the plane coordinate. By mapping to the plane, the operator $O$ (now inserted at $z = e^{\frac{\beta}{2}\frac{\theta}{2\pi}}$) picks up a factor of $e^{\frac{\beta}{2}\frac{\theta}{2\pi}\Delta_O}$ due to the conformal transformation.

2. Now, we can insert an energy eigenstate on one end by gluing in a unit disk with an operator $O_i$ inserted at the origin. However, note that we cannot simply insert the state along the other circle, since it is currently located at $|z| = e^{\beta/2}$ rather than the unit circle. In order to glue the other state, we first perform an overall rescaling by $e^{-\beta/2}$. The operators $O$ and $O_i$ transform under this rescaling and give a total contribution of $e^{-\frac{\beta}{2}(\Delta_{O_i}+\Delta_O)}$. We can now glue the state on the other circle by inserting the complement of the unit disk with an operator $O_j$ inserted at infinity.

3. We now have a three-point function on the plane and we would like to relate it to an OPE coefficient. To do so, we must have the three operators at $0, 1$ and $\infty$. Operators $O_i$ and $O_j$ are already located at the appropriate positions but $O$ is not. We therefore need to perform an extra rescaling by $e^{\frac{\beta}{2}-\frac{\theta}{2\pi}\frac{\beta}{2}}$. This will give a total contribution of $e^{(\frac{\beta}{2}-\frac{\theta}{2\pi}\frac{\beta}{2})(\Delta_i-\Delta_j+\Delta_O)}$. Note that the scaling contribution coming from the operator at infinity is negative because the state is defined as $\lim_{z\to\infty}\langle 0|z^{2\Delta_j}O_j(z)$ and therefore carries effective weight $-\Delta_j$.

Putting everything together and restoring energies on the cylinder rather than conformal dimensions, we find

$$c_{ij} = C_{Oij}e^{\frac{\beta}{2}\frac{\theta}{2\pi}(E_j-E_i)-\frac{\beta}{2}E_j}, \tag{3.24}$$

which means the normalized extended Hilbert space state is

$$|\psi_E(O)\rangle = \frac{1}{\sqrt{\mathcal{N}(O,\beta)}}\sum_{i,j}C_{Oij}e^{\frac{\beta}{2}\frac{\theta}{2\pi}(E_j-E_i)-\frac{\beta}{2}E_j}|E_i\rangle \otimes |E_j\rangle, \tag{3.25}$$

where we have defined the normalization as

$$\mathcal{N}(O,\beta) = \sum_{i,j}|C_{Oij}|^2 e^{\beta\frac{\theta}{2\pi}(E_j-E_i)-\beta E_j}. \tag{3.26}$$

Note that the normalization factor is a torus two-point function. The state reduces to the usual thermofield-double state when there is no operator inserted (namely when $O$ is the identity $O = \mathbb{1}$). It is also symmetric under the exchange of $i$ and $j$ and a simultaneous transformation $\theta \to 2\pi - \theta$, as expected by the symmetries of the problem.

We can now immediately compute the reduced density matrix for our interval $A$:

$$\rho_A = \frac{1}{\mathcal{N}(O,\beta)}\sum_{ijk}C_{Oij}C_{Oik}e^{\frac{\beta}{2}\frac{\theta}{2\pi}(E_j+E_k-2E_i)-\frac{\beta}{2}(E_j+E_k)}|E_j\rangle\langle E_k|, \tag{3.27}$$

from which we could compute its von Neumann entropy. This however requires diagonalizing the matrix (3.27), which is complicated. We will therefore perform the replica trick instead and compute the Rényi entropies. We would like to emphasize that the problem is not conceptual, and that we have the direct Hilbert space expression for the reduced density matrix. One could work with this object directly, and diagonalization is possible in certain limits like a small interval expansion. We simply chose to do the replica trick in order to show a general matching with the CFT answer, (2.14).

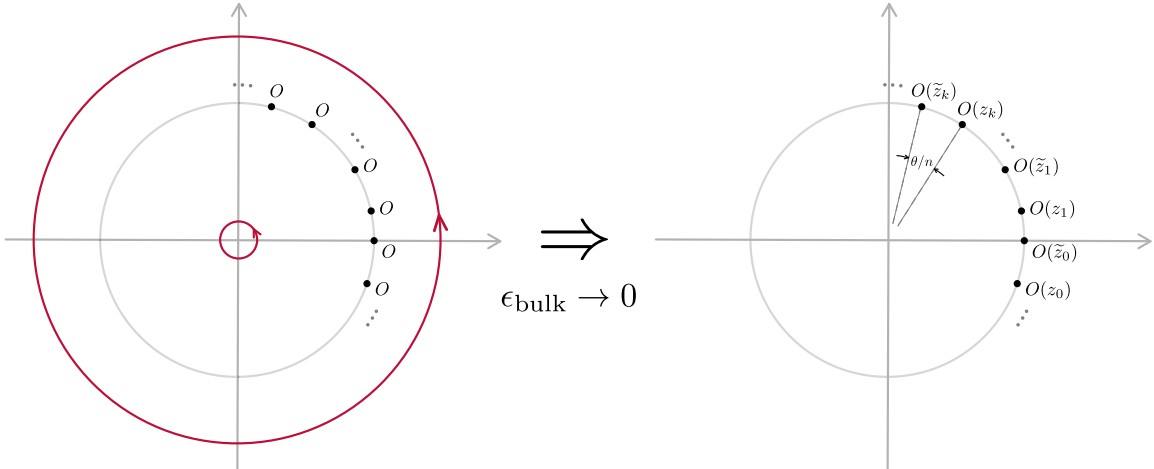

Figure 6: *Left*: The bulk geometry at finite $\beta$. The inner red circle is the tube around the entanglement cut and is identified with the outer red circle. The operator insertions are indicated with black dots. *Right*: The bulk geometry in the limit $\beta \to 0$ or equivalently $\epsilon_{\text{bulk}} \to 0$. The inner circle has now shrunk to a point and the geometry has become the two-dimensional plane. The operators are now situated along the unit circle. The coordinates $z_i$ and $\tilde{z}_i$ are given in (2.10).

**The Rényi entropies for the excited state**

We now wish to compute the Rényi entropies for the excited state (3.27). Just like we did in the holographic CFT, we will compute the different of Rényi entropies between the excited state and the vacuum . We have

$$\Delta S_n = \frac{1}{1-n} \log \text{Tr} \frac{\rho_A^n}{\rho_{A,\text{vac}}^n}, \tag{3.28}$$

with

$$\rho_{A,\text{vac}}(\beta) = \frac{1}{Z(\beta)} \sum_i e^{-\beta E_i} |\psi_i\rangle \langle\psi_i|, \tag{3.29}$$

the vacuum density matrix. Note that $\text{Tr}\rho_A^n$ is, up to the normalization factors which are themselves 2-point functions, a $2n$-point correlation function on a torus of length $2\pi n$, see figure 6. We now rewrite the difference of Rényi entropies as

$$\Delta S_n = \frac{1}{1-n} \log \left[ \frac{\langle O_1 ... O_{2n} \rangle_{n\beta}}{Z(n\beta)} \left( \frac{Z(\beta)}{\mathcal{N}(O, \beta)} \right)^n \right]. \tag{3.30}$$

For simplicity we have dropped the insertion points of the operators. Next, we perform an $S$-transformation and find

$$
\begin{aligned}
\Delta S_n &= \frac{1}{1-n} \log \left[ \frac{\langle O_1 ... O_{2n} \rangle_{\frac{4\pi^2}{n\beta}}}{Z(\frac{4\pi^2}{n\beta})} \left( \frac{4\pi^2}{n\beta} \right)^{2nh} \left( \frac{Z(\frac{4\pi^2}{\beta})}{\mathcal{N}(O, \frac{4\pi^2}{\beta})} \right)^n \left( \frac{4\pi^2}{\beta} \right)^{-2nh} \right] \\
&= \frac{1}{1-n} \log \left[ n^{-2nh} \frac{\langle O_1 ... O_{2n} \rangle_{\frac{4\pi^2}{n\beta}}}{Z(\frac{4\pi^2}{n\beta})} \left( \frac{Z(\frac{4\pi^2}{\beta})}{\mathcal{N}(O, \frac{4\pi^2}{\beta})} \right)^n \right].
\end{aligned}
\tag{3.31}
$$

Now, we can take the cutoff to zero, which from (3.11) means we take $\beta \to 0$. In this limit, the torus extends into a cylinder; in figure 6 the inner circle shrinks and the (identified) outer

circle expands. Thus the correlation functions become vacuum correlation functions with the operators located exactly as in the original 2d CFT calculation in figure 2, and we find

$$\Delta S_n = \frac{1}{1-n} \log n^{-2nh} \frac{\langle O_1...O_{2n}\rangle}{(\langle O_1 O_2\rangle)^n}. \tag{3.32}$$

Since the operators we are considering is the current operator, we can compute the correlation function exactly and we find

$$\Delta S_n = \frac{1}{1-n} \log\left(\left(\frac{1}{n}\sin\left[\frac{\theta}{2}\right]\right)^{2n} \mathrm{Hf}(M_{ij})\right), \tag{3.33}$$

which is in complete agreement with the CFT answer (2.11). One can also perform the analytic continuation and obtain the entanglement entropy, which again will match the CFT answer.

A few comments are in order. First, note that both the bulk entanglement entropy computation and the boundary CFT entanglement entropy computation are performed by computations in a 2d CFT. Note however that the 2d CFTs are different! The boundary CFT is a holographic large $c$ CFT, while the bulk computation involves the conformal field theory living at the boundary of a Chern-Simons theory, in this case a chiral boson theory. As a consequence, the bulk entanglement entropy answer (3.33) is exact. On the contrary, the large $c$ CFT answer (2.11) may in general not be exact. For scalar excitations, it is not. It can receive additional $1/c$ corrections coming from the interactions within the matter sector or with gravitons. Nevertheless, for our particular state which is a current insertion, both the holographic CFT answer and the bulk entanglement entropy answers are exact, since large $N$ factorization is exact for $U(1)$ currents.

# 4 Bulk entanglement entropy for gravitons

The computations we presented in the previous section apply much more generally than just Abelian Chern-Simons theories. The non-Abelian generalisation is straightfoward and in particular, we can in principle immediately apply our formalism to AdS$_3$ gravity.

The bulk effective field theory we will consider is now a rewriting of the Einstein-Hilbert action as a topological gauge theory: an $SO(2,1) \times SO(2,1)$ CS theory [33, 64–66]. Just as with $U(1)$ CS theory, the only non-trivial degrees of freedom live on the boundary and in this case are large diffeomorphisms that don't vanish sufficiently quickly at the boundary. These are known as boundary gravitons [32]. Since the degrees of freedom lie entirely at the boundary, one can formulate a purely boundary description of these degrees of freedom. In [33] this boundary theory was derived from the $SO(2,1) \times SO(2,1)$ CS theory by considering particular (AdS) boundary conditions for the CS gauge fields. They found that the Euclidean boundary action is $S = S_+[\phi] + S_-[\bar{\phi}]$, where

$$S_+[\phi] = \frac{c}{24\pi} \int d^2x \left(\frac{\phi''\overline{\partial}\phi'}{\phi'^2} - \phi'\overline{\partial}\phi\right), \tag{4.1}$$

and analogously for $S_-$ with $\partial$ instead of $\overline{\partial}$. The primes indicated derivatives along the contractible cycle (in the bulk). Here $c = 3\ell_{\mathrm{AdS}}/2G_N$. Crucially, this theory is not modular invariant as it just keeps track of the fluctuations around a particular saddle. As we will see, for us, the thermal AdS$_3$ saddle will be of most importance, in which case $\phi$ and $\bar{\phi}$ take values in $\mathrm{Diff}(S^1)/PSL(2,\mathbb{R})$. When dimensionally reducing to a single boundary dimension, (4.1) becomes the Schwarzian theory and so (4.1) can be thought of as the three dimensional analogue of it.

To compute the bulk entanglement entropy, we will employ the same type of regulator as for the $U(1)$ case described in the previous section and illustrated in figure 3. We again pick boundary conditions such that the edge degrees of freedom are the same as the boundary degrees of freedom, with a transparent boundary condition at the interface.

We stress here that we have not explicitly checked that this is always possible; a subtlety could arise from the fact that the entanglement cut is not an asymptotic boundary like that of AdS, and therefore may not have a Fefferman-Graham expansion, (the Chern-Simons analogue of) which was an important technical tool in the construction of [33]. We expect however that the construction should still be possible as the bulk theory is topological and does not depend on a choice of metric.

As in the $U(1)$ case, this edge theory is not modular invariant, thus we need to specify exactly *which* partition function we are interested in computing. This ambiguity is fixed by bulk data; from the bulk geometry in figure 3 it is clear that in our set up the spatial circle is contractible, and thus we should consider a thermal AdS$_3$ saddle. (In the $U(1)$ case, we enforced that the spatial circle be similarly contractible by demanding that there be no Wilson lines in the interior; choosing the thermal AdS$_3$ saddle is the gravitational analogue of that choice). Moreover, given that we are interested in the $\beta \to 0$ limit, the thermal AdS$_3$ will be very hot.

For the bulk entanglement entropy, we therefore need to compute the high temperature torus partition function of the boundary modes. This is given by the vacuum character for the Virasoro descendants 3.13 [33]

$$\chi_{\text{vac}}(\beta) = e^{K\beta} \prod_{n \geq 2} \frac{1}{(1 - e^{-\beta n})^2}, \tag{4.2}$$

where $K$ is the Casimir energy in the vacuum due to the boundary graviton excitations. The value of $K$ does not affect our results; in particular, an overall multiplicative factor of the form $e^{K\beta}$ in the partition function $Z(\beta)$ never contributes to the entropy. [8]

As $\beta \to 0$ it is instead the asymptotic growth of the Virasoro descendants in (4.2) that will contribute and not the Casimir energy piece. We can directly extract the asymptotic behavior from (4.2) to be:

$$\chi_{\text{vac}}(\beta) \sim e^{\frac{\pi^2}{3\beta}}. \tag{4.3}$$

The $\beta \to 0$ limit giving us a very hot thermal AdS$_3$ should be contrasted with the typical situation, where the dominant gravitational saddle – the BTZ black hole at high temperature – is considered. Since the boundary theory we consider is not modular invariant, the asymptotic growth of these two does not agree, indeed it is shown in [33] that $\chi_{\text{BTZ}}(\beta) \sim e^{\frac{4\pi^2 K}{\beta}}$.

From (4.3) we can directly compute the bulk entanglement entropy. We find

$$S_{\text{bulk}} = \frac{1}{3} \log\left( \frac{2}{\epsilon_{\text{CFT}}} \sin\left( \frac{\theta}{2} \right) \right) + \frac{1}{3} \frac{L}{2\epsilon_{\text{bulk}}}. \tag{4.4}$$

It is tempting to interpret the coefficient of the logarithmic term here to constitute a shift of the holographic central charge by *one*, as expected as we are counting virasoro descendants. This interpretation is actually somewhat clouded, as for a holographic interpretation we must take $L$ to be the length of the bulk geodesic; we can now no longer consider both terms separately, and we find:

$$S_{\text{bulk}} = \left( \frac{\ell_{\text{AdS}}}{2\epsilon_{\text{bulk}}} + \frac{1}{3} \right) \log\left( \frac{2}{\epsilon_{\text{CFT}}} \sin\left( \frac{\theta}{2} \right) \right). \tag{4.5}$$

---

[8]One simple way to see this is to note that such a factor corresponds to shifting all energies by a constant, which clearly has no information-theoretical content. In theories enjoying modular invariance the ground-state energy can be indirectly related to the high-energy density of states, but the boundary graviton theory in question is not modular invariant.

We now see that it is difficult to disentangle a finite shift from the UV divergent contribution. We comment on this further in the conclusion.

# 5 Discussion

In this paper, we considered the bulk entanglement entropy of gauge fields and gravitons in AdS$_3$/CFT$_2$. The topological nature of the bulk EFT enabled us to explicitly compute the bulk entanglement entropy. We picked a regulator such that the dynamics on the entanglement cut are the same as the AdS boundary, along with a transparent boundary condition on the cut. Given this choice of regulator, we presented a concrete map to the extended Hilbert space, which is a finite temperature regularization. We were able to compute the bulk entanglement entropy in the vacuum for both gauge fields and gravitons, and found

$$S_{\text{CFT}} = \frac{A}{4G_N} + S_{\text{bulk}} = \left( \frac{\ell_{\text{AdS}}}{2G_N} + \frac{c_{\text{top}}}{3} \frac{\ell_{\text{AdS}}}{\epsilon_{\text{bulk}}} + \frac{c_{\text{top}}}{3} \right) \log \left( \frac{2}{\epsilon_{\text{CFT}}} \sin \left( \frac{\theta}{2} \right) \right), \qquad (5.1)$$

where $c_{\text{top}}$ counts the number of (boundary) degrees of freedom of the bulk effective field theory. For the photons, we also considered excited states given by acting with the current on the vacuum. The map to the extended Hilbert space yields a generalization of the thermofield-double state, created by performing the euclidean path integral with an operator inserted. We computed the difference of the bulk entanglement entropy between the excited state and the vacuum. We found a perfect match with the CFT computation, providing a non-trivial check of the FLM formula. There are only very few explicit tests of the FLM formula and they either compute quantities fixed by conformal symmetry [67] or require small interval expansions [41, 68]. To the best of our knowledge, our result is the only test of the FLM formula that works for arbitrary interval size; it is however still somewhat kinematical in that the holographic dictionary does not involve dynamical bulk degrees of freedom.

In the remaining of this section, we discuss open questions and further directions.

**Universality of constant terms in the bulk entanglement entropy?**

While the entanglement entropy is often divergent and therefore regulator dependent, it is known that certain terms in the cut-off expansion are universal and carry physical information (for example they can serve as monotonic $c$-functions). An example of such quantities are the coefficients of the log terms in even dimensions which encode the $A$-type anomaly. In odd dimensions, the universal terms are not related to an anomaly but can may also be related to monotonic functions along RG flows. Such an example is the constant term in a three-dimensional theory for the entanglement entropy of a disk, obtained from the mutual information of concentric disks [69].

Our set-up studies the entanglement entropy of a three-dimensional bulk theory and it is therefore interesting to understand whether certain terms we computed are universal and carry physical information. The first type of term we encountered was a constant term in the $U(1)$ Chern-Simons entanglement entropy (3.17)

$$S_{\text{top}} = \log S_{00}. \qquad (5.2)$$

This is the usual topological entanglement entropy, and in a generic gapped theory one can construct an entanglement geometry to extract this term.[9] In our set-up where we consider Chern-Simons theory on a spatial disk with the entanglement cut intersecting the boundary

---

[9]See e.g. the continuum discussion of [27], who study the entanglement entropy of Chern-Simons theory on a spatial sphere, where the entangling surface is the hemisphere.

circle, such a constant term is modified by rescaling the boundary cutoff; in other words, the topological term is contaminated with gapless modes arising from the boundary circle, and our setup is not well-suited for extracting the topological entanglement entropy.

We turn now to the gravitational interpretation of our results; examining the structure of (5.1) it is tempting to interpret the divergent piece of our answer as a renormalization of Newton's constant of the form:

$$\frac{1}{4G_N} \to \frac{1}{4G_N} + \frac{c_{\text{top}}}{12}\frac{1}{\epsilon_{\text{bulk}}}, \tag{5.3}$$

together with a modification of the relationship between the holographic central charge and (the running) Newton's constant:

$$c = \frac{3\ell_{\text{AdS}}}{2G_N} + c_{\text{top}}. \tag{5.4}$$

Written this way one is tempted to conclude that $c_{\text{top}}$ is universal. This is somewhat dangerous, as we point out by noting a curiosity in the gravitational case, where in the scheme above we found that $c_{\text{top}} = 1$ from counting Virasoro descendants. Note that the value of $K$ in (4.2) – i.e. the contribution to the Casimir energy from graviton fluctuations – can also be understood as a shift of the holographic central charge from the Brown-Henneaux value. Interestingly, a direct computation of $K$ in [33,70] computed within zeta function regularization correspond to a shift of $c$ by 13, and not 1. (At a calculational level this can be traced back to the lack of modular invariance of the virasoro character (4.2).)[10]

This cannot be considered a discrepancy, as the separation advocated in (5.3) and (5.4) cannot be made in a universal manner; there is no simple way to compare the regulator used in our computations (the radius of a small tube cut out from the bulk) from the zeta function regulator of [33,70]. It would be very interesting to find a way to compare these regulators (or find a different way to give universal meaning to $G_N$ and $c$ separately). On general grounds, it would be interesting to understand whether there exists a universal term in the entanglement entropy of three-dimensional theories for a spatial disk split in half. This would provide a path towards understanding the universality of the constant $c_{\text{top}}$.

## Excited boundary graviton states

Just as for the photons, we can also consider excitations of the boundary gravitons. In particular, the states one would like to consider are then of the form $T(0)|0\rangle$.[11] We would have to consider a $2n$-point function of stress tensors to compute the entanglement entropy of this state relative to the vacuum. Although such correlators can be computed recursively, the resulting form is not nicely expressible in terms of a Haffnian, because the stress-tensor is not a Virasoro primary and there will be additional terms coming from the Schwarzian. Furthermore, it is not known how to proceed with the analytic continuation in this case. Similar conclusions also hold for non-Abelian (compact) gauge groups. In that case one would consider Kac-Moody current states $J^a(0)|0\rangle$ labelled by an Lie algebra index $a$. The $2n$-point functions are again fixed by symmetry, but due to non-trivial tensor structure, they are still complicated. We leave a more detailed study of these excited states for both compact and non-compact non-Abelian gauge groups and their entanglement entropy to future work.

---

[10]Note that the quantum corrections to the CFT entanglement entropy were computed in [71] using the replica trick rather than the FLM formula, which yields a shift of the central charge by 13. Such a computation boils down to evaluating the sphere partition function, which is UV-regulated the same way as in [33,70]. It is therefore expected that it picks up a shift by 13.

[11]We are grateful to Sagar Lokhande for early discussion and collaboration on this point.

$CFT_E$ **vs.** $CFT_B$

We emphasize that in principle, there are two "skin" CFTs in our setup: one living on the entanglement cut , $\text{CFT}_E$, and one living on the physical AdS boundary, $\text{CFT}_B$. Their properties arise from distinct physical considerations: $\text{CFT}_E$ arises from the properties of the UV cutoff regulating the bulk topological theory, whereas $\text{CFT}_B$ arises from the boundary conditions at the AdS boundary. Thus in principle they might be different. If they are distinct, then a full specification of the problem requires both a description of the two CFTs, as well as a description of how they join together. The universal data characterizing this joining is that of a conformal interface between the two CFTs. We discuss a simple example where $\text{CFT}_E$ and $\text{CFT}_B$ can differ by by considering a doubled Chern-Simons theory (whose edge modes form a *non*-chiral boson) in Appendix A.2.

In fact, in certain cases it is possible to gap out $\text{CFT}_E$ entirely. Whether or not this is possible is equivalent to asking whether a general topological phase can admit a gapped boundary. The general case of a collection of Abelian gauge fields $A_I$ has been discussed in [24, 25]. This depends on algebraic properties of the matrix $K_{IJ}$ coupling together the Chern-Simons gauge fields. If $\text{CFT}_E$ is gapped then rather than specify an interface between $\text{CFT}_E$ and $\text{CFT}_B$ we should simply specify a boundary state that terminates $\text{CFT}_B$; in this case the region of the CFT torus that results in the $\frac{L}{\epsilon_{\text{bulk}}}$ contribution would vanish. (This of course does not mean that the entanglement entropy is finite; there will be other non-universal contributions that do not arise from the considerations of this paper).

Finally, we find it interesting that in the AdS/CFT context, this suggests that the set of possible ways to factorize the Hilbert space at an entanglement cut in three-dimensional quantum gravity can be understood by classifying all possible boundary states and conformal interfaces in the edge theory of [33]. We feel this deserves further study.

# Acknowledgments

We are grateful to Maissam Barkeshli, Jan de Boer, Diptarka Das, Thomas Dumitrescu, Steve Giddings, Diego Hofman, Kristan Jensen, Sagar Lokhande, Marcos Marino, Edward Mazenc, Onkar Parrikar, Gabor Sarosi, Ronak Soni and Simon Ross for helpful discussions. AB is partly supported by the NWO VENI grant 680-47-464 / 4114. JK is supported by the Simons Foundation. NI is supported in part by the STFC under consolidated grant ST/L000407/1.

# A   Known facts about Abelian Chern-Simons theories

In this appendix we review some features of Abelian Chern-Simons theories. See, for instance [58, 59, 72] for more details.

## A.1   $U(1)$ **chiral Chern-Simons theory**

On a manifold $M$ with boundary $\partial M$, the $U(1)$ Chern-Simons action

$$S_{CS}[A] = \frac{k}{4\pi} \int A \wedge dA, \tag{A.1}$$

must be supplemented with a boundary term in order to obtain a well-defined variational principle. Assuming the boundary to be flat 2d space labeled by complex coordinates $(z, \bar{z})$,

we take this boundary term to be:

$$S[A] = S_{CS}[A] + \frac{k}{4\pi} \int_{\partial \mathcal{M}} dz d\bar{z} A_z A_{\bar{z}}, \tag{A.2}$$

such that the on-shell variation of the total action is

$$\delta S[A] = \frac{k}{2\pi} \int_{\partial \mathcal{M}} dz d\bar{z} A_z \delta A_{\bar{z}}. \tag{A.3}$$

Thus $A_{\bar{z}}$ is the source and $\frac{k}{2\pi} A_z$ is the response. For a well-defined variational principle we now demand $\delta A_{\bar{z}} = 0$.

The combined action (A.2) defines a chiral boson living on the boundary. To understand this, consider a fully gauge-fixed potential $\bar{A}$ in the bulk so that $A = \bar{A} + d\phi$, with $\phi$ a gauge parameter. (A.2) is not quite gauge-invariant:

$$S[\bar{A} + d\phi] = S[\bar{A}] + \frac{k}{2\pi} \int_{\partial \mathcal{M}} dz d\bar{z} \left( \bar{A}_{\bar{z}} \partial \phi + \frac{1}{2} \partial \phi \bar{\partial} \phi \right). \tag{A.4}$$

Thus we see that the putative gauge mode $\phi$ has acquired dynamics on the boundary, i.e. the 2d kinetic term associated with a scalar field. The boundary condition translates into $\bar{\partial}\phi = 0$. Our normalization of this kinetic term is non-standard; this is because the periodicity of the boson is always $2\pi$ from the compactness condition (3.3). The bulk CS action has a boundary mode that is a *chiral* boson, whose holomorphic part couples to the external source $\bar{A}_{\bar{z}}$ in precisely the expected manner. The conventionally normalized Kac-Moody current is given by

$$j = k \partial \phi. \tag{A.5}$$

A more careful canonical quantization of Chern-Simons theory on $D^2 \times \mathbb{R}$ gives the same result [58]. Perturbative excitations of $\phi$ map to the modes of the Kac-Moody current via (A.5).

The boson $\phi$ is periodic with period $2\pi$; we may thus also consider states where $\phi$ winds around the boundary. In particular, consider an excitation where $\phi$ winds through $2\pi$; from (A.5) this maps to a state with $U(1)$ charge $k$. In the language of the edge boson, this state is created by the chiral vertex operator $e^{ik\phi(z)}$. Note that in this state $\phi$ winds around a cycle that shrinks in the bulk, and thus there will be a bulk radius where $A \sim d\phi$ naively becomes ill-defined. This corresponds to a properly quantized (and thus unobservable) Dirac string carrying $2\pi$ flux, and as usual should be considered non-singular. It is an allowable state in the spectrum.

This can be contrasted with states where $\phi$ winds instead only through a fraction of its full range; from the point of view of the $U(1)$ charge algebra, they are constructed by operators of the form $e^{im\phi}$, $m \in \{1, \cdots k-1\}$. Interpreted in terms of gauge field data alone these states are indeed singular: rather they correspond to the insertion of a bulk Wilson line. This bulk Wilson line is necessarily characterized by extra data (i.e. the mass of the associated particle, etc.), presumably arising from the UV completion of the bulk theory. One can also show that from their coupling to the bulk Chern-Simons field they acquire an anomalous dimension $h_{\text{anom}} = \frac{m^2}{2k}$, and thus are generically non-local operators unless this fractional spin is compensated by some extra dynamics (e.g. a coupling to an *anti*-holomorphic gauge field, etc.).

Groups of $k$ of these minimally charged Wilson lines can combine on a bulk magnetic monopole and vanish; in the chiral operator language, this means that the product of $k$ minimal vertex operators fuses to give a charge $k$ *chiral* vertex operator, which is now a part of the symmetry algebra. In other words, the $k$-fold Wilson line is now a descendant.

## A.2 Doubled Chern-Simons theory

Let us now consider a Chern-Simons action consisting of two gauge fields $B$ and $C$:

$$S[B,C] = \frac{k}{2\pi} \int B \wedge dC. \tag{A.6}$$

Unlike (3.2), this theory is parity-invariant. It is roughly equivalent to two copies of (3.2) with opposite level. Holographically it contains both holomorphic and anti-holomorphic currents that can be assembled into a single $U(1)$ current $j$.

Just as above, this bulk action must be supplemented with a boundary action; here we follow the discussion of [73], who studied similar theories in a higher-dimensional context. Consider the following boundary term:

$$S = S[B,C] + \frac{g^2}{2} \frac{k}{2\pi} \int_{\partial \mathcal{M}} B \wedge \star_2 B. \tag{A.7}$$

Here $g$ is a non-universal parameter that is associated with the choice of boundary conditions. Now an on-shell variation of the action results in

$$\delta S = -\frac{k}{2\pi} \int_{\partial \mathcal{M}} B \wedge \left( C - g^2 \star_2 \delta B \right). \tag{A.8}$$

This prompts us to identify a current $j$ and a source $a$ as

$$j = \frac{k}{2\pi} \star_2 B, \qquad a = C - g^2 \star_2 B. \tag{A.9}$$

Note that here both of the two components of $j$ are independent, and it contains both holomorphic and antiholomorphic pieces. In the absence of the source $a = 0$, the bulk equations of motion $dB = dC = 0$ imply

$$d \star_2 j = dj = 0, \tag{A.10}$$

which we recognize as the simultaneous conservation of the axial and vector currents (or, equivalently, holomorphic and anti-holomorphic) currents. These simultaneous conservation equations actually immediately imply the existence of a boundary boson [74, 75]; to make manifest its emergence from bulk gauge redundancy, it is again instructive to write $B = \overline{B} + d\phi$, with $\phi$ a compact gauge parameters. We then find, analogously to (A.4):

$$S = S[\overline{B}, C] + \frac{k}{2\pi} \int_{\partial \mathcal{M}} \left( \frac{g^2}{2} d\phi \wedge \star_2 d\phi + a \wedge d\phi \right). \tag{A.11}$$

Here $\phi$ is unconstrained, and it is thus a regular *non*-chiral boson propagating on the boundary. Here the current $j$ is

$$j = \frac{k}{2\pi} \star d\phi. \tag{A.12}$$

Note that here the radius of the boson can be tuned by continuously adjusting the parameter $g$. In fact, this theory can be gapped out entirely by adding terms of the form $\cos(k\phi)$ to the boundary action. Our discussion of the Chern-Simons theory did not contain any such free non-universal parameters; this arose as we insisted on boundary Lorenz invariance. In applications to the FQHE boundary Lorentz invariance is relaxed and the velocity of the boundary mode is then also adjustable.

Using this doubled Chern-Simons theory, it is easy to understand different choices for $\mathrm{CFT}_E$ and $\mathrm{CFT}_B$. As we showed in the above, the boundary theory is that of a single unconstrained

compact scalar $\phi$ with period $2\pi$. Here $g$ is a non-universal parameter – the boson radius – that can be freely adjusted. Thus it is perfectly possible to imagine (e.g.) that $CFT_E$ has a value $g_E$ and $CFT_B$ has a distinct value $g_B$; this is a particularly simple case where the two CFTs are different in that they describe a non-chiral boson with distinct radii.

We briefly sketch how the computation of the vacuum entanglement entropy would proceed in such a case. We are now computing the entanglement entropy on a torus that is made of two annular regions joined together; one annular region is inhabited with a boson with a radius $g_E$, and the other annuluar region is inhabited by a boson with radius $g_B$. The two annuli are joined by a radius-changing conformal interface, as described in [76]. As the central charge is the same on both sides of the interface, it is easy to see that the Cardy limit of the partition function on the torus will still take the form (3.17), except with an extra non-universal contribution that is a function of $g_E, g_B$. It would be interesting if such a contribution could be extracted in a numerical study of a system defined on the lattice.

# B Entanglement entropy computation from wave-functions in Chern-Simons theory

Let us study the entanglement entropy of the vacuum. In this case, we are interested in computing the partition function of Chern-Simons theory on a solid torus, whose boundary is a $\mathbb{T}^2$ with modular parameter $\tau$,

$$\tau = \frac{i\pi}{\log\left(\frac{2}{\epsilon_F}\sin\left(\frac{\theta}{2}\right)\right)}. \tag{B.1}$$

Let us begin by studying the wavefunctional for $U(1)$ Chern-Simons theory on a solid torus. Besides the dependence on $\tau$ and the gauge field on the boundary torus, these wavefunctionals also depend on a label $r$. This label is associated to the number of Wilson lines inserted in the bulk and takes values between 0 and $k$. From the boundary point of view it labels the different irreps of the extended $\mathfrak{u}(1)$ Kac-Moody algebra at level $k$. To write the wavefunctionals, we choose complex coordinates $z$ on the torus and choose $A_{\bar{z}}$ as our coordinate of the wavefunction. We will furthermore decompose the gauge field on the torus as

$$A_{\bar{z}} = \partial_{\bar{z}}\chi + \frac{i\pi}{\tau_2}a. \tag{B.2}$$

In terms of this data, the wavefunctionals are given by [77],

$$\Psi_r[A_{\bar{z}};\tau] = \frac{1}{\eta(\tau)}\vartheta\begin{bmatrix}r/k\\0\end{bmatrix}(ka|k\tau)\exp\left(\frac{k\pi}{2\tau_2}a^2\right)\exp\left(\frac{ik}{4\pi}\int d^2z\,\partial_z\chi\,\partial_{\bar{z}}\chi\right). \tag{B.3}$$

To compute the vacuum entanglement entropy, we only need this wavefunctional at the origin, $\Psi_r[0;\tau]$. An immediate problem here is that in our set-up $\tau \to i0^+$ and since such limits of theta functions are a bit tricky, we will rewrite $\Psi_r[0;\tau]$, using standard transformation rules of the theta and dedekind eta functions, as

$$\Psi_r[0;\tau] = \frac{1}{\sqrt{k}\eta(-1/\tau)}\vartheta\begin{bmatrix}0\\0\end{bmatrix}\left(\frac{r}{k}\,\middle|\,\frac{-1}{k\tau}\right). \tag{B.4}$$

Notice that the wavefunctional is only modular invariant for $k=1$ and $r=0$. The entanglement entropy computation now amounts to computing,

$$S_{EE}^{\text{vac}} = \lim_{n\to 1}\frac{1}{1-n}\log\left(\frac{\Psi_r[0,n\tau]}{\Psi_r[0,\tau]^n}\right). \tag{B.5}$$

Using that as $\text{Im}(\tau) \to 0^+$,

$$\eta(-1/\tau) \to e^{-\frac{\pi i}{12\tau}}, \quad \vartheta\begin{bmatrix} 0 \\ 0 \end{bmatrix}\left(\frac{r}{k}\,\middle|\,\frac{-1}{k\tau}\right) \to 1, \tag{B.6}$$

we find that the entanglement entropy of the vacuum is given by

$$S_{EE}^{\text{vac}} = \frac{1}{6}\log\left(\frac{2}{\epsilon_F}\sin\left(\frac{\theta}{2}\right)\right) - \frac{1}{2}\log k. \tag{B.7}$$

The first term in this expression is the usual expression that we expect for a $c = 1/2$ conformal field theory. The second piece is the topological entanglement entropy and is independent of what type $r$ of anyon is inserted in the bulk. This is to be expected, since $S_{\text{top}} = \log(S_0^r)$ for a type $r$ anyon and $S_0^r = 1/\sqrt{k}$ for the chiral free boson, which is indeed independent of $r$.

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
