# Peer review of "Bulk entanglement entropy for photons and gravitons in AdS$_3$"

_SciPost Physics, doi:SciPost Phys. 8, 075 (2020)_

## Round 2 · Referee Report · Anonymous (Referee 1) · 2020-2-3

Strengths

1) Exceptionally well written

2) Relevant topic, combines multiple strands of research

3) Reaches an intriguing result and corroborates the HLM prescription

Weaknesses

1) A few physical points were addressed in insufficient detail - see Report

Report

This paper was a pleasure to read. Overall it is written very clearly, which makes it very easy to go through and understand. Using rather standard techniques for calculating thermal partition functions of 2D CFTs, the authors calculated entanglement entropies of topological theories in $AdS_3$ for subregions that contain a piece of the AdS boundary (more precisely, for bulk entanglement wedges corresponding to boundary regions). Several examples were studied: vacua of $U(1)$ Chern-Simons theories, simple excited states of $U(1)$ Chern-Simons theories, and vacua of Einstein gravity (which, in $AdS_3$, is a Chern-Simons theory with a non-compact gauge group). The result indicated that the universal piece of entanglement entropy takes the form characteristic for 2D theories, though with bulk and boundary data contributing parametrically different terms. No (obvious) 3D-specific universal terms were found.

The calculation was very concrete, presented succinctly but not sloppily, and I believe it sets a good standard for the field.

I have a few comments or confusions. They can be divided into three classes, according to severity:

1) Very important conceptual issue:

When studying $U(1)$ gauge fields in $AdS_3$, the authors work with a pure Chern-Simons theory. This is a good warm-up for studying pure gravity, and this choice can be picked by fiat too. But I am less comfortable with this calculation if Chern-Simons is an effective bulk theory (as mentioned at the top of page 3). The reason is that entanglement entropy has the pesky ability to sense UV physics (see Kabat, Shenker, Strassler [hep-th/9506182]). In the context of 3D gauge theories, this Maxwell vs Chern-Simons issue was studied by Agon, Headrick, Jafferis and Kasko [arXiv:1310.4886] and by Radicevic [arXiv:1509.08478]. These papers all claim that, depending on the size of the region $L$, the universal term will either be of the form $\frac12 \log k$ or $\frac12 \log(e^2L)$, where $e$ is the Maxwell coupling. The crossover between these universal terms happens at $L \sim k/e^2$. In particular, if $k \rightarrow \infty$, as the authors assume, then the Maxwell-appropriate universal term will always be in the entropy, even though Chern-Simons dominates the IR physics at the level of correlation functions.

Of course, all these results are in flat space. In AdS, the size of the region $L$ diverges, so maybe you can pick a regime where $L \gg k/e^2$. This may be impossible if $k \sim c_{\textrm{CFT}}$, as mentioned in footnote 3. Maybe a way out is to find that the entire crossover doesn't happen in AdS. That would be an interesting finding on its own; I am not aware that anyone checked this. I think there is some interesting physics here. At worst, the authors can explicitly restrict to studying pure (not effective) Chern-Simons theories in the bulk.

2) Important clarifications:

2.1) I understand the authors take large $k$ because that is featured in brane constructions. But is that necessary for this calculation, assuming the Chern-Simons theory is pure? Maybe the assumption can be dropped altogether?

2.2) In Sec. 3.1, I would like to ask the authors to consistently use terms "states" and "operators". The state-operator correspondence does not work in the bulk, so the sentence "In particular, states..." turned out slightly confusing. I think this sentence meant "The bulk is topologically trivial, so Wilson lines in the bulk are trivial, so their boundary duals are trivial too, and hence the boundary states these boundary operators correspond to are not in the Hilbert space of interest." I would also delete references to dualities of oranges...

2.3) The conformal mapping in Sec 3.3 is stated a bit unclearly. The sentence "this surface is topologically a torus" seems to be referring to the bulk tube that was just defined - and topologically this is manifestly a cylinder, not a torus. I am pretty sure the authors were imagining a torus obtained by concatenating the bulk tube/cylinder with a corresponding boundary tube, but I admit I found this hard to infer solely from the text or from Fig. 4. I think it would be very helpful to indicate $\epsilon_F$ on Fig. 3, and perhaps to very explicitly state which surfaces we are talking about before ever conformally mapping them.

3) Minor kvetching:

3.1) It's Kac (or Кац in Russian), not Kač (or Кач). A lot of the old literature mysteriously changes his name, as was done 4 out of 9 times in this paper.

3.2) It's Virasoro, not Virasoso (2 out of 7 times in this paper)

3.3) I understand the authors meant "boundary photons" when choosing the title, but I wonder if it can be changed to more generically refer to gauge fields or gauge theories, especially in view of my comment 1). Chern-Simons doesn't really have photons on its own. Same for gravitons.

3.4) Have the authors considered comparing their result to the bulk entropy associated to an "island" in the bulk, i.e. to a region that doesn't extend all the way to the boundary? It would be simple to calculate but may be nice to present.

3.5) Speaking of universal terms on subregions with boundaries, have the authors found any inspiration in the vast literature on entropic g-functions? This was done mostly for conformal theories (Jensen and O'Bannon arXiv:1509.02160 seems like a relevant starting point for 3D theories). It would at least be nice to contrast those results with the ones obtained here.

3.6) In equation (1.5), perhaps $A$ should have been replaced by $L$?

3.7) What is the boundary dual of $\epsilon_{\textrm{bulk}}$? Less annoyingly said, how should it scale with other "small" parameters, like $\epsilon_{\textrm{CFT}}$, $G_N$...?

3.8) What symmetry are we talking about when we talk about symmetry-breaking relevant deformations at the bottom of page 10?

3.9) Fig. 3 is very nice, but for a bit I was confused by the gray line that almost (but not quite) overlaps with the red line $\gamma_A$. Maybe that gray line can be removed?

---

## Round 3 · Author Response

Dear editor,

We thank the referee for their careful reading and valuable and much appreciated comments, questions and suggestions for improvement. We reply to them in turn below.

1) We thank the referee for this very interesting and important question. We agree that from an effective field theory description the Maxwell term will generically exist. However let us denote by L the (proper) bulk size of interest; though k may be very large, we are actually always interested in a regime where L ≫ k/e2. The reason is that L is a distance to the AdS boundary; thus its size is controlled by the CFT UV cutoff, which is chosen (independent of k) such that L ≫ k/e2 (though k is large too). In particular, the UV-cutoff is not constrained by the central charge of the CFT. Thus for the purposes of this problem we believe it suffices to focus purely on the Chern-Simons term.
We agree with the referee that understanding whether the transition of the two regimes happens for a spherical region in AdS is an interesting question in its own right, and would be worth investigating in the future. We have added a comment about this on p9.

2.1) Taking k large is not necessary at all for the purposes of our calculation. We simply mean that the application to holography would imply a large value of k. We have changed the sentence before the footnote accordingly.

2.2) We thank the referee for this comment and we have clarified this sentence and added a footnote in the revised manuscript. We would like to emphasize that the state operator correspondence does work in the bulk: Any state of the bulk Chern-Simons theory is given by the insertion of the appropriate operator at the south pole of the boundary sphere. Concerning the metaphor of oranges, we actually do think it is helpful (e.g. see the next point).

2.3) The bulk tube cuts out two circles at the boundary sphere and so the resulting surface is a torus. It is like cutting a cylindrical hole through an orange, resulting in a surface that is topologically a solid torus.

3.1,3.2) We thank the referee for spotting these typos. We have changed them in the revised manuscript.

3.3) We thank the referee for the feedback on the title, but we feel it is clear from the abstract what is meant with the title and so we refrain from any changes to the current title.

3.4) This is a very interesting comment, but we have not considered such ’islands’ in this manuscript. This also connects to the question raised before about the Maxwell term. This would be the appropriate way to address that question, namely by considering islands.

3.5) That is a great question and certainly worth investigating, but we have not done so in this manuscript.

3.6) The first equality is just a restatement of the FLM relation (given also in 1.1), which is then worked out the second equality. We therefore believe it is not needed to replace A with L there.

3.7) That is a (yet another) great question. It is important to emphasize that the CFT answer must be blind to ε_bulk: It cannot know about the effective bulk cut- off introduced, since by definition it only has access to bulk UV-finite quantities. Formulating a way to understand what a bulk cutoff is in the CFT (for example an upper bound on scaling dimensions of operators) is an important question in AdS/CFT that remains unanswered. From the bulk side, it would seem natural to pick L_planck ≪ ε_bulk ≪ L_AdS but it was not really relevant in our computations.

3.8) We had in mind specific examples where the UV regulator breaks a shift symmetry whereas the edge mode does not; however the idea is more general than this example, and so we have simply removed the words “symmetry-breaking.”

3.9) We thank the referee for this comment. The figure has been changed.

We hope that this answers the questions and comments raised by the referee.

Sincerely yours,
Alexandre Belin, Nabil Iqbal and Jorrit Kruthoff

Resubmission 1912.00024v3 on 10 April 2020
Submission 1912.00024v2 on 24 January 2020

---

## Editorial Decision

published